# An approach for evaluating the role of protection measures in rock fall hazard zoning based on the Swiss experience

Erika Prina Howald[1], Jacopo Maria Abbruzzese[1], Chiara Grisanti[1]

[1] Territorial Engineering Institute (Insit), School of Engineering and Management Vaud (HEIG-VD), Yverdon-les-bains, 1401, Switzerland

*Correspondence to*: Jacopo M. Abbruzzese (jacopo.abbruzzese@heig-vd.ch)

**Abstract.** Rock fall hazard zoning is essential for ensuring the safety of communities settled at the toe of potentially unstable slopes. Rock fall hazard zoning can be performed including the effect of protection measures, when land use restrictions might not be enough to mitigate hazards. The real effectiveness of the measures must be assessed to make sure they can play their role, especially in those cases when measures might have been installed at a given site since years. This article focuses on how to evaluate the effectiveness of rock fall protection measures and how hazard zoning can be influenced by their correct operation. The approach presented is divided into four main stages, which include a two-step procedure to evaluate the effectiveness of both existing and new protections. It is based on quite a comprehensive rock fall protection database built for the Canton of Vaud in Switzerland, and on the Swiss Federal Guidelines for hazard zoning; however, all the methodological framework proposed and related considerations could be in principle extended to any other regional or national contexts in which a combination of intensity and frequency is used to assess rock fall hazards.

## 1 Introduction

Mountainous regions all over the world are affected by rock fall hazards, which may constitute a serious threat to the safety of local communities settled at the toe of rocky slopes. Especially in the last decades, the need for new areas to be exploited for urban development caused human settlements to be located even in potential rock fall prone areas. National authorities have been trying to establish appropriate hazard and risk management policies to cope with landslides, including rock falls (Cascini et al., 2005; Labiouse and Abbruzzese, 2011).

Some countries were actually able to define a comprehensive set of guidelines for landslide hazard management at the national (or at least regional) level (AGS 2002; Fell et al., 2008; Raetzo et al., 2002; M.A.T.E./M.E.T.L., 1999; Copons et al., 2003). Switzerland is indeed among these countries; a framework for coping with rock falls as well as all other types of landslides exists for the whole Confederation (Raetzo et al., 2002; Lateltin et al., 2005). In the Swiss codes, the priority in trying to reduce potential risks associated to rock falls is given to appropriate land use planning. Hazard zoning maps are elaborated for delineating areas of conflict between human assets and the rock fall runout, and each level of hazard in the maps corresponds

to constraints to the construction, set for each of the hazard levels defined in the guidelines, both for new and already existing urban areas.

The principle of trying to manage rock fall hazards with an appropriate use of the territory implies that the use of protection measures is not necessarily considered at first. However, if land use regulations alone can be very effective in reducing the

exposure to hazards in areas destined to new urban development, protection measures still have to be used to keep existing built areas safe, should these areas be located in zones where hazard is found to be not negligible. Land use planning therefore has eventually been coupled quite often with protection measures all over the Swiss territory.

As rock fall protections have been installed throughout the last decades, their design and installation have complied to different standards, as long as hazard management policies, hazard modelling procedures and technical progress concerning the

protections design advanced. With particular regards to the existing protection measures, therefore, the problem arises whether they can comply to the current standards and should be accounted for in hazard zoning; furthermore, if this is the case, the next point to tackle is how this should be done. As a function of their current state, the perfect functionality of some existing measures might in fact be not guaranteed; consequently, their performance capabilities should be ascertained, in order to understand if they can still play their role and possibly influence hazard zoning and therefore land use planning (and, if so, up

to which extent).

Similar considerations on the actual effectiveness of a protection measure not only concern existing measures but could in fact be extended even to new protections, for which other types of factors, not strictly related to the protection itself, could still reduce their capabilities, despite they are newly designed (for instance environmental factors/conditions characterising the site where the measures are located).

The way to take existing and new protection measures into account in hazard analyses still remains an open question and requires further development.

In this context, the work presented in this paper aims at proposing a methodological framework for evaluating the role of rock fall protections (both existing and new) on hazard assessment at a given site. The approach was developed at the Territorial Engineering Institute (Insit) of the University of applied sciences and arts of Western Switzerland (HEIG-VD), Yverdon-les-

bains. It is based on a methodology developed by the Institute initially for the Canton of Vaud, with the objective of establishing a simple yet effective procedure for evaluating the conditions and effectiveness of the protection measures on its territory.

After briefly recalling the principles of rock fall hazard zoning in Switzerland, the next Sections illustrate the details of the approach proposed and some recommendations on its application.

**2 Landslide hazard zoning in Switzerland**

Landslide hazard can be defined as a condition which may adversely affect human life, property or activity to the extent of causing disasters. Hazard description must then account for intensity and frequency of occurrence of the dangerous phenomenon within a given period of time (Fell et al., 2005; Fell et al., 2008).

In line with this definition, the Swiss Codes (Raetzo et al., 2002; Lateltin et al., 2005) characterise landslide hazards based on a combination of intensity and frequency of occurrence. Specifically regarding rock falls, these two parameters are represented, respectively, by rock fall energy and return period, i.e. inverse of the frequency of occurrence (given by the product of the frequency of failure of a rock compartment and frequency of reach of the detached block). Energy and return period are both classified into three categories (low, moderate, high) and their combination defines three levels of hazard: low, moderate and

high, as graphically explained by the matrix diagram showed in Fig.1. An additional hazard level qualified as "residual" is defined for return periods higher than 300 years. Differently from the meaning of residual hazard most commonly adopted, usually interpreted as the hazard still affecting a site after all efforts to mitigate it have been made, in the Swiss Codes (and throughout this paper) the residual hazard is intended as the possible occurrence of extreme rare events (but no further specification or well defined classes are given in terms of return period and energy to classify this hazard).

As shown in the matrix diagram, each hazard level is associated to a specific colour: red for high, blue for moderate, yellow for low, and yellow-white hatched for residual hazard, respectively. As briefly mentioned in the Introduction, these four colours not only classify the severity of potential rock fall hazards in the zoning maps: in fact, they are also associated to the specific land use planning regulations corresponding to each level of hazard in the intensity-frequency diagram, for both new areas to be built and already existing settlements.

The constraints set for each degree of hazard for regulating land use can be summarised as follows and as reported in Fig. 1:

- the red zone (high hazard) is a "prohibition domain": new constructions and further developments of the existing areas are forbidden;

- the blue zone (moderate hazard) is a "prescription domain": new constructions and further development of built
areas are allowed under conditions;

- the yellow zone is an "awareness domain": communities are informed of potential low hazard but urban development is, in principle, allowed.

- The yellow-white hatched zone is also an "awareness domain", the difference with the yellow being that in this area the hazard is known to be only at a residual level (in the sense specified above according to the Swiss Codes).

Once validated, rock fall hazard zoning maps are integrated into land use planning and become legally binding.

## 3 Approach for evaluating the effectiveness of protection measures

Many different types of rock fall protection measures exist (examples in Fig. 2) and they do interact with the process in very different ways. Some guidelines exist on the criteria behind an appropriate rock fall protection strategy (Volkwein et al., 2011; CCC, 2013) and on the correct design of several measures available for this purpose (Volkwein et al., 2005; Muhuntham et al., 2005).

According to the guidelines and principles included in the Swiss strategy for coping with natural hazards, the reliability of protection measures depends on (i) the type of measure, (ii) their proper design and (iii) their current conditions in terms of possible flaws or shortcomings (Keusen et al., 2008). The first two criteria concern both new and existing measures. On the other hand, the current conditions of the protection are a crucial point to consider particularly for protection measures which are already existing at a given site. Indeed, it is very important to assess their actual state before relying on their design capacities, as, in fact:

- despite the measure has already been installed for a relatively long time frame (and maybe even impacted by some previous events), it could still work as expected, so that in principle no reason would lead to believe it should no longer be serviceable;
- on the contrary, the measure might actually be partially or even totally unable to play its role because of, for instance, partial damages due to previous events, lack of maintenance of other type, etc.

Based on these considerations, an approach for establishing how well protection measures, either new or existing, can actually work would be quite useful, in order to establish what changes they generate in the hazard assessed at a given site and, ultimately, how land use planning changes at the site. On the other hand, a methodology which might go further than qualitative considerations like those contained in the Swiss guidelines (Keusen et al., 2008) is still required, for delineating how this kind of evaluation of the measures effectiveness should be carried out.

In the attempt to provide a solution to the problem of assessing the state of the measures and how they can perform (and possibly modify the hazard zoning that would be assessed without them), a procedure is proposed, which aims to take into account those factors which can influence the behaviour of a rock fall protection and evaluate the effectiveness of the protection accordingly. This approach includes:

1. the elaboration of a database, which was built to have a record of all the types of protection measures used in the Canton of Vaud, to know what types of protections are used, what their technical specifications are and therefore also what their flaws and shortcomings can be, based on issues linked to the design, operation, installation, maintenance, environment in which they are installed etc.;

2. a methodology, articulated into four steps, whose aim is indeed to evaluate what is the actual effectiveness of a rock fall protection and, ultimately, what the influence of the presence of the protection is on hazard zoning and land use planning.

The next Sections give details on the database and other input data for the application of the methodology, as well as on each of the four steps of the proposed evaluation procedure.

### 3.1 Inventory of the existing protection measures

Some inventories for protection measures already exist in Switzerland (Canton of Grisons, 2017; Frei, 2013; BAFU, 2017); for this work, a database of all the types of protection measures specifically present on the territory of the Canton of Vaud was elaborated (Grisanti and Prina Howald, 2014, 2015a and 2015b), based on information and data collected on site, provided by the authorities of the Canton and the companies producing the protections used on the territory.

The database was implemented in Microsoft Access and contains quite detailed information about each type of protection. In particular, it counts 23 types, classified into 7 categories: barrier fences, dams, wire mesh/cable nets, walls, topographic modifications (slope re-profiling), anchors and protection galleries.

An Access form is dedicated to each measure, and relevant data are stored in 33 fields, which synthesise information on several aspects including the operation of the protection, technical specifications (capacity, height, etc.), drawbacks, norms to be respected for their design, installation, costs, a list of potential flaws and shortcomings, principles/needs for maintenance and reparation. An example of form for a given type of barrier fence is showed in Fig. 3 (vertical barrier fence in single line, low energy absorption capacity).

On the one hand, when new protections need to be installed, this tool can assist the choice of an appropriate measure, by comparing several types of protections and analysing features, advantages and possible drawbacks in relation to the problem studied (e.g. hazard scenario, environment, etc.). At the same time, it defines most of the technical aspects which influence the behaviour of the protection measures during their life span, which are very important when existing measures have to be inspected for evaluating their conditions and, consequently, effectiveness.

### 3.2. Methodology

The objective of this procedure, based on the Swiss Codes, is to provide a simple yet effective tool for analysing whether the measure can play the role it was designed for, in order to lower the hazard at a given area, and, if so, eventually reclassifying the hazard and requalifying the area in terms of land use.

The methodology is characterised by the following four steps (Fig. 4), detailed in the next Sections:

- characterisation of hazard and evaluation of the existing measures/choice of new measures;
- analysis of the measures according to "Scenario 0": effective capacity of the protection;
- analysis of the measures according to "Scenarios 1 to 6": reduced capacity of the protection;
- reclassification of hazard by using the intensity-frequency diagram (in case the measure properly mitigates the hazard according to design).

### 3.2.1 Characterisation of hazard and evaluation of the current state of the protection measures

The input data for the hazard analysis in presence of protection measures are:

- information about the site and characterisation of the hazardous event, in terms of volume, size of the block(s), energy and probability of occurrence;
- information about the protection measures to be built or already existing.

Relevant information about the site and the event are in principle available from the hazard analysis previously carried out at the considered site for a given hazard scenario, and the corresponding zoning map (which of course at this stage does not account for any possible existing protection). This allows to characterise the hazard all over the slope and to represent on the Swiss intensity-frequency diagram the initial scenario at any point of the slope (Fig. 5).

Then, an investigation on the type of protection measures existing, if any, is carried out, or, if protection is required and no measures have been installed yet, an appropriate choice for a newly designed one has to be made. Regarding the information for evaluating the conditions of the existing protection measures, the fields and attributes defined in the database presented in Section 3.1 constitute a fairly comprehensive description of each protection and help in establishing which data is required to be collected. In particular, the correct operation of a rock fall protection can be influenced by a number of factors which were grouped as follows:

1) factors linked to the particular environmental conditions characterising the site;
2) factors associated to the general rock fall protection design and possible additional engineering solutions to optimise the design;
3) factors specifically linked to the flaws and drawbacks of each type of rock fall protection.

On this basis, forms for collecting data related to these factors were proposed not only to gather all the relevant information needed, but also to do it in a structured way. This last aspect can help in uniforming the procedure of data collection and achieving a certain degree of consistency and reproducibility of the data collection phase. The forms are composed of three sections, each dedicated to one of the aspects above mentioned (site, type and peculiar designing features of the protection measures, flaws of the protection measures).

The next Sections present how these factors are taken into account in the elaboration of a two-step methodology for evaluating the effectiveness of the protections.

### 3.2.2 Evaluation Step 1 – Analysis of the measures according to "Scenario 0": effective capacity of the protections

The first stage of evaluation proposed in this methodology takes into account only factors linked to the site (presence of protection forest, rivers, freezing/thawing cycles, snowfall, outcropping rock, other natural hazards, damages due to animals, etc.) and the "general" design of the protection measures installed (manufacturing faults, possibility of plastic deformations,

respect of the norms, homogeneity/weak points in the structure, coupling with other protection measures, redundancy of main structural elements, features related to creep failure, etc.) - i.e. factors 1) and 2) in the previous Section.

In other words, at this stage the initial condition of the measure is supposed to be that of new protection, working and interacting with the environment as well as potential events according to its nominal capacity (i.e. the capacity it was designed for). This condition was named "Scenario 0".

The objective of this first analysis is however to define whether from this condition the nominal capacity of the rock fall protection might nevertheless be lowered by any of the factors considered. If so, this decrease must be evaluated, and an "effective capacity" has to be computed and used in place of the nominal capacity for the next steps of the methodology.

Ideally, this can be achieved in a relatively simple way, using a heuristic approach and defining at first, for each factor ($j$) belonging to groups 1 or 2, "penalty coefficients" $P_{01,(j)}$, $P_{02,(j)}$ (where 0 refers to Scenario 0, while 1 and 2 refer to factor types 1 and 2, respectively). These penalty coefficients can then be applied to the capacity of the protection (e.g. energy absorption capacity of a barrier fence) to obtain a lower capacity than the nominal.

For instance, the presence of other natural hazards (factor type 1) can partially damage a barrier fence (or a dam) and cause a loss of energy absorption capacity. The effective capacity of the barrier (or dam) can be thus computed as:

$$E_{eff} = E_{opt} \cdot P_{01,(j)} \qquad\qquad (1)$$

where $E_{eff}$ is the effective capacity (in this case expressed in terms of energy absorption) of the barrier fence, $E_{opt}$ its nominal capacity and $P_{01,(j)}$ the penalty coefficient associated to the factor $j$= "other natural hazards", belonging to group 1.

This step is applied to any case study, involving either new or existing protection measures. In the first case, it allows to consider which factors could lower the capacity of a measure despite it is newly designed; in the second, this evaluation constitutes the starting point for the following step, when flaws and shortcomings of existing measures are also taken into account.

### 3.2.3 Evaluation Step 2 – Analysis of the measures according to "Scenarios 1 to 6": reduced capacity of the protection

This second step of the methodology was specifically designed for existing protection measures and focuses on evaluating the consequences of flaws and shortcomings affecting each specific type. Based on the types of measures considered in the database and on the data collected on each of them, potential causes of malfunctioning were classified into six categories, common for all the protection measures. Each category of flaws was associated to a new "scenario" of analysis. These conditions are defined as "Scenario 1", "Scenario 2"… "Scenario 6". The types of faults and associated scenarios were established as follows:

- issues related to the positioning of the protection measure (Scenario 1);
- problems due to a non-correct design of the measure (Scenario 2);

- faults and problems related to the construction and/or installation of the measure (Scenario 3);

- flaws/faults due to lack of maintenance (Scenario 4);

- shortcomings due to the fact that the measure has attained its life span (Scenario 5);

- the measure is working in residual conditions, i.e. beyond its life span (Scenario 6)

As done in the previous phase, the objective of this one is also to determine how much the capacity of a protection measure can be lowered. This time, the source of capacity reduction is on the other hand due to the fact that the protection might have been installed for a long time, might have interacted with the environment and, possibly, even one or more previous rock fall events, or might have been not maintained as required, etc.

Starting from the effective capacity $E_{eff}$ calculated at the previous stage, and based on the same principle, penalty coefficients
can be proposed to evaluate to which extent the capacity of the existing protection is reduced. If for a given type of measure $P_{i,(j)}$ indicates the penalty coefficient associated to the $j$ factor of Scenario $i$ (i=1..6), the "reduced capacity" of a protection can be determined by means of an expression similar to Eq. (1). With reference to the example of a barrier fence, a factor like a damage to the supports of the barrier after a previous impact (Scenario 4) can lower the capacity of the barrier, as the net will not have the necessary support to retain the energy it was designed for. Therefore, its reduced capacity (again expressed in
terms of absorption energy) can be evaluated as:

$$E_{red} = E_{eff} \cdot P_{4,(j)} \qquad\qquad (2)$$

where $E_{red}$ is the reduced capacity of the barrier fence, $E_{eff}$ is the effective capacity and $P_{4,(j)}$ the penalty coefficient associated
to the factor $j=$ "damage to supports", belonging to category of faults n.4, i.e. Scenario 4.

For every Scenario, therefore, all those factors actually affecting the protection at the site have to be assigned proper corresponding penalty coefficients. These coefficients can then all be applied to the effective capacity, for evaluating the reduced capacity.

Contrary to the factors considered in the previous step, which often might modify the capacity of the measure to a lesser extent,
the factors related to Scenarios 1 to 6 could actually prove the protection measure to be totally unserviceable, as a function of its current state.

### 3.2.4 Reclassification of hazard

The two step evaluation explained in the previous Sections allows for computing the actual capacity (effective and reduced) of the protection measure considered, starting from the nominal. As a result, by comparing at the location of the protection the
hazard affecting the slope (in absence of measures) to the capacity obtained, it can be established whether the protection can actually act against the event and mitigate the hazard beyond that location. In particular, the following possibilities could happen:

a) the capacity of the measure is not sufficient to lower the hazard; the protection measure could be destroyed by the event (e.g. energy absorption capacity of a barrier fence lower than the energy of the event). The protection measure cannot be taken into account in its state:

- for *existing measures*: the protection should be: (i) replaced for ensuring the proper level of safety for the assets it was meant to protect when it was first installed; (ii) ignored (for what concerns its effect), if transformations in terms of land use have already modified the assets at risk and no properties are highly exposed right beyond the protection.

- for *new measures*: the design solution should be (i) modified/upgraded with additional solutions which can avoid the effect of factors 1) and 2) (Sections 3.2.1 and 3.2.2); (ii) changed to another type of measures - especially if the new choice of the measure is less sensitive to factors 1) and 2).

b) The capacity of the protection is sufficient for mitigating the hazard, but it might not guarantee a totally satisfactory margin for safety (for instance, the energy absorption capacity of e.g. a barrier fence, or a dam, stands fairly or too close to the energy of the event). In these cases, further considerations on the specific problem can lead to:

- for *existing measures*: decision to (i) repair/upgrade substantially or (ii) replace the measure, similarly to what discussed for the previous situation (point a), as a function of the type/amount of flaws. For instance, maintenance and/or partial reparation could be enough in some cases, but replacement might be required if flaws and partial damages cannot be handled otherwise;

- for *new measures*: (i) modify or (ii) change the project, as discussed for the previous situation - point a);

c) the capacity can still be fairly sufficient to mitigate the hazard (e.g. energy absorbed and height of a barrier fence); the protection can be considered in the hazard assessment, most probably only minor interventions in terms of maintenance are required.

In situations b) and c) described above, the role of the protection can actually lower the values of rock fall energy and/or return period obtained without protections, in all areas affected by the process runout and located beyond the measure. A reclassification of hazard can be then performed at the locations concerned by using the Swiss intensity-frequency diagram. In particular, the position of the energy-return period couple corresponding to the original hazard scenario, i.e. "natural" scenario (Point A, Fig. 6), can be shifted to another area of the diagram, as a function of the new energy-return period couple which can be determined after the hazard has been re-assessed accounting for protection measures. New hazard levels can be determined based on the effect of the measure: if this effect is such that the point representing the natural scenario is shifted to an area of the diagram corresponding to a lower hazard (Point B in Fig. 6), the area concerned can be assigned that (lower) level of hazard. Land use regulations will therefore be defined accordingly, based on the new hazard level determined.

However, to clearly distinguish between the situations in which a e.g. "low" degree of hazard derives from the absence of a substantial hazard and the situation in which "low" hazard is obtained thanks to the performance of protection measures, all areas concerned by a change in hazard due to the role of the protection measures will be marked on the zoning maps as hatched

zones. The hatched textures will show both the colour corresponding to the hazard level before the installation of the protection and the colour corresponding to the new hazard level obtained. Such representation system sums up the fact that if, on the one hand, the hazard level at the area concerned is the one expressed by the colour corresponding to the lower level in the hatched texture, this level has been achieved, on the other hand, only because of the correct performance of a protection measure on site.

## 4 Scheme of application of the methodology

In this Section, principles of application of the methodology are presented separately, with some further details and accompanied by a theoretical example, for each of the two situations in which (i) new protection measures have to be installed, or (ii) the presence of already existing measures has to be considered. As mentioned before, for both types of analyses, the rock fall hazard affecting the study site is preliminarily known.

### 4.1 New protection measures

The elaboration of a rock fall hazard zoning map at a given site might show that some assets could be located in areas with too high rock fall potential risks. The areas concerned require therefore protection, to be ensured by newly designed measures. In such a case, the methodology can be applied as follows. Referring to Fig.7:

1) *Choice of the protection measure(s)*. The first point to tackle is the choice of the appropriate measure to be installed:

    1.a)       consideration of the hazard scenario;

    1.b)       exam of the environmental conditions of the site

    1.c)       choice of the appropriate protection measure: the database presented in Section 3.1 can assist the choice of the engineer also in this sense, showing which protections are more frequently used for a given situation; more than one option can initially be considered and/or a combination of more measures (each of them will be evaluated in the next step of the procedure);

2) *Evaluation of the effectiveness according to Scenario 0*. Evaluation of the performance of the selected measure according to Scenario 0:

    2.a)       analysis of the factors belonging to group 1) (Section 3.2.1) which can affect the protection in the problem examined;

    2.b)       analysis of the factors belonging to group 2) (Section 3.2.1);

    2.c)       definition of penalty coefficients for factors of types 1) and 2);

    2.d)       computation of the effective capacity of the protection by means of Eq. (1)

    If the reduction of capacity is negligible or low, only smaller (or no) interventions could be necessary for the measure in order to avoid this reduction, and have the measure working according to its nominal specifications. For instance, if

aggressive water is present at the site and might generate corrosion and problems to the foundations of a barrier fence, drainage systems and appropriate treatment of the metal components against corrosion should be adopted, together with as much appropriate maintenance operations, to avoid a loss of efficiency of the protection in time.

2.e)        if more than one possible protection measure was evaluated, steps 2.a) to 2.d) should be repeated for all the protections considered, so that a final choice for the hazard mitigation can be taken, based on how much the factors considered can potentially lower the capacity of the measure, and therefore what modifications and/or additional solutions have to be considered in terms of engineering design to avoid this (design, installation, costs, operation, performance, maintenance…)

3) *Reclassification of hazard and requalification of the areas concerned in terms of land use.* Reclassification of hazard and requalification of affected areas are technically possible, as the good performance of the protection has been guaranteed by its correct design and possible complimentary solutions adopted, according to the previous steps of evaluation. The reclassification of the hazard levels affecting the areas located beyond the protection can then be carried out as pointed out in Section 3.2.4, at all locations concerned by the rock fall runout.

### 4.1.1 Example 1

The following example (as well as the one in Section 4.2.1) aims at explaining how to apply the methodology in practice to a given study case. However, as some methodological aspects and the testing phase of the whole procedure described are still under development, the example(s) is(are) purely theoretical and based on assumption which might help in giving a clear view on how to perform each step of the procedure, rather than focusing on quantitative results.

Let the slope represented in Fig. 9 be characterised by the presence of a rock fall prone area at its top; a few buildings (e.g. chalets, holiday resorts, etc.) are situated downslope, at a given location x. The buildings are potentially threatened by a hazard scenario given by one block falling on average every 60 years; the hazard associated to this scenario has been evaluated based on trajectory simulation results and would provide the information and zoning map schematically shown in Fig. 9. The built area is situated within the moderate hazard zone, which would require local protection of all the buildings involved (Fig.1). On the other hand, the installation of protection measures at an appropriate location along the slope might potentially represent the optimal solution for ensuring the safety of the elements exposed, and lower the hazard at the endangered area. The methodology described above can then be applied for investigating this possibility and its steps can be summed up as follows.

1) *Choice of the protection measure(s).*

   1.a) The hazard threatening the area was characterised as:

      - 1 block of 1.2 m$^3$ falling on average every T=60 years;
      - maximum energy at the area of the buildings: 150 kJ

   1.b)  Peculiar conditions of the site are assumed to be: (i) the frequent presence of snow and (ii) the presence of debris flows in the same area

1.c) Due to several types of constraints linked to the project and the study area, low energy barrier fences are selected to be designed. According to the database presented in Section 3.1, these barriers can retain energy up to 750 kJ. As the maximum energy of the event at the location of the barrier fence is 400 kJ (Fig. 9) and the rebound height 2 m, the design specifications for the barrier considered in this example will be, e.g., an energy absorption capacity of 500 kJ and a height of 3 m.

2) *Evaluation of the effectiveness according to Scenario 0.*

2.a) In this example, as the barrier fence will be newly installed, the factors considered in Scenario 0 should in principle not reduce the capacity of the barrier, since they should be already accounted for in its design standards and recommendations (and therefore, possible appropriate engineering solutions should already be taken at this stage). However, it will be assumed that, as it may happen, some complexities and peculiarities of the examined problem and related uncertainties (e.g. the occurrence of the debris flows which may contribute to damage the barrier), justify the use of penalty coefficients $P_{01,(snow)}$ and $P_{01,(debris)}$. This allows to consider a sort of "most unfavourable condition" in which snow and potential solid material from the debris flow might partially fill up the nets of the barrier fence, reducing their capacity to stop the blocks.

2.b) The appropriate design of the barrier makes all factors in group 2 negligible in the problem examined (new barrier, i.e. no damages and/or issues due to any previous use).

2.c) It is assumed that, after an appropriate calibration of the penalty coefficients, the following values (indicative) are obtained:

$P_{01,(snow)} = 0.98$

$P_{01,(debris)} = 0.92$

The effect of having the nets of the barrier fences partially filled up essentially influences the probability of occurrence, as filled nets reduce the effective height of the barriers (i.e. more blocks could potentially travel beyond the barrier). A relatively simple idea for calibrating factors reducing the height of the barrier could be based on the results of rock fall simulations, run either with barriers of reduced height, according to the expected volumes filling up the net, or with a slightly modified topographic profile (and related physical properties) just before the barrier, to simulate the pile of material filling the nets. By checking how many blocks travel beyond the barrier fence in these conditions, compared to the number of blocks travelling beyond the barrier in perfect working order, one can establish what the increase in the probability of reach is in case the nets are filled and, therefore, what penalty coefficient to apply, depending on this increase (e.g. if in the case of partially filled nets 10% of the total simulated blocks travel beyond a barrier, which in perfect working order stops 100% of the blocks, the penalty coefficient could be set to 0.90).

2.d) Assuming that the penalty coefficients modify mostly the probability of reach, the associated change in the return period of the event at the locations beyond the barrier can be computed as:

$$T_{eff} = T_{opt} \cdot P_{01,(snow)} \cdot P_{01,(debris)} =$$
$$= T_{opt} \cdot 0.98 \cdot 0.92 = T_{opt} \cdot 0.90$$

The effective capacity of the barrier fence installed therefore is:

$$E_{eff} \cong E_{opt} = 500 \text{ kJ}$$
$$T_{eff} = T_{opt} \cdot 0.90$$

For a probability of detachment of the blocks equal to 1/60 years = 0.0167 at the source area, and considering that the probability of reach $P_r$ of the blocks at the location $x_p$ of the barrier is 0.75 (Fig. 9), the probability of occurrence $P$ at $x_p$ before the measure is installed is $P(x_p) = 0.0167 \cdot 0.75$ years$^{-1}$, i.e. the return period at this location is $T_0 = 1/P(x_p) = 80$ years. If the trajectory simulation results obtained with the barrier in place yield that the barrier fence in perfect working order stops 50% of the blocks which reach its location, the probability of occurrence at $x_p$ goes down to $P'(x_p) = 0.0167 \cdot 0.75 \cdot 0.5 = 6.26 \cdot 10^{-3}$ years$^{-1}$, i.e. $T_{opt} = 1/P'(x_p) = 160$ years. Finally, taking the penalty coefficients into account, the value of the "effective" return period at $x_p$ is:

$$T_{eff} = T_{opt} \cdot 0.90 = 160 \cdot 0.9 = 144 \cong 140 \text{ years}$$

To a first approximation, by applying the same reduction to all locations beyond the barrier, the effective return period can be estimated with the same procedure. At the location $x^*$ of the buildings, for a probability of reach without barrier equal to 0.68 (Fig. 9), it results:

- return period without barrier fence:

$$T_0 = 1/(0.0167 \cdot 0.68) = 88 \cong 90 \text{ years}$$

- return period with barrier fence:

$$T_{opt} = 1/(0.0167 \cdot 0.68 \cdot 0.5) = 176 \cong 180 \text{ years}$$

- "effective" value of return period with barrier fence:

$$T_{eff} = T_{opt} \cdot 0.90 = 180 \cdot 0.9 = 162 \cong 160 \text{ years}$$

If based on the trajectory results obtained the energy at the location of the buildings in presence of a perfectly working newly designed barrier fence is 20 kJ (Fig. 9), the new rock fall hazard at the location x is given by:

$$E(x) = 20 \text{ kJ}$$
$$T(x) = T_{eff}(x) = T_{opt}(x) \cdot 0.90 = 160 \text{ years}$$

3) *Reclassification of hazard and requalification of the areas concerned in terms of land use.*

A reclassification of hazard can in fact be established for the built area, which will not be any more characterised by a moderate hazard but will pass to low hazard (Fig. 9). In particular, the couple ($T$, $E$) representing the initial hazard (moderate) will move downwards diagonally to the point representing the new hazard level (low), as the effect of a reduction of both energy and return period at the location of interest. A requalification in terms of land use of this area can be considered, i.e., in this example, further urban development would be allowed with no restrictions (Fig.1). Similarly, by repeating this procedure for all the locations beyond the barrier, the new boundaries between the hazard zones in presence of protection measures can be delineated.

## 4.2 Existing protection measures

When protection measures exist at a given site and their effectiveness has to be evaluated, the application of the methodology includes (Fig. 8):

1) *Analysis of the conditions of the existing measure(s).*

 1.a) consideration of the hazard scenario;

 1.b) exam of the environmental conditions of the site

 1.c) collection of all data and information concerning the type of measures existing at the study site and their current state (forms described in Section 3.2.1)

2) *Evaluation of the effectiveness according to Scenario 0.* The analysis performed at this stage is basically the same as the one described for new protections:

 2.a) analysis of the factors belonging to group 1);

 2.b) analysis of the factors belonging to group 2);

 2.c) definition of penalty coefficients for factors of types 1) and 2);

 2.d) computation of the effective capacity of the protection by means of Eq. (1)

 2.e) if more than one measure exists at the site, steps 2.a) to 2.d) should be repeated for each measure.

3) *Evaluation of the effectiveness according to Scenarios 1 to 6.* The effect of flaws and shortcomings of the protection measure(s) on its (their) performance are determined at this stage:

 3.a) analysis of which factors associated to Scenarios 1 to 6 which can affect the protection in the problem examined;

 3.b) definition of penalty coefficients for all these factors (Scenarios 1 to 6);

 3.c) computation of the reduced capacity of the protection by means of Eq. (2)

If the effective capacity is only slightly inferior to the nominal capacity, most probably repairing/replacing some parts which might have been damaged from previous smaller rock fall events or other causes can be sufficient to have the protection restored to good working order.

3.d) if more than one protection measure is present at the site and need to be investigated, steps 3.a) to 3.c) should be repeated for all the protections;

3.e) Design of new protections, if existing ones are not adequate for hazard mitigation, according to the procedure in Section 4.1.

4) *Reclassification of hazard and requalification of the areas concerned in terms of land use.* Contrary to the case of new measures, where the reclassification of hazard could in principle always be done, for existing measures this can happen only when, even in presence of faults and flaws, the effective capacity is still such that the measure can cope with the event. If this is the case, the reclassification can be done, again, as described in Section 3.2.4, at all locations concerned by the rock fall runout.

## 4.2.1 Example 2

The example (Fig. 9) resumes the hazard scenario and set up of the example in Section 4.1.1, the only difference being that the barrier fence considered in this example is an *already existing* barrier, which was designed with a nominal energy absorption capacity of 500 kJ.

1) *Analysis of the conditions of the existing measure(s).*

Steps 1.a) and 1.b) remain the same as in Section 4.1.1. On the other hand:

1.c) the collection of information and data concerning the current state of the barrier fence (Section 3.2.1) evidences that some blocks coming from previous events were not removed from the nets, and that some of them caused some damage to the nets and the posts.

2) *Evaluation of the effectiveness according to Scenario 0.*

As obtained from the previous example:

$E_{eff} \cong E_{opt} = 500 \text{ kJ}$

$T_{eff} = T_{opt} \cdot 0.90$

3) *Evaluation of the effectiveness according to Scenarios 1 to 6.*

3.a) From the data collected, the factors associated to Scenarios 1 to 6 are essentially linked to maintenance problems (Scenario 4). Among these, two factors defined for barrier fences can be selected, which correspond to "damages to the posts/nets after impacts" ($P_{4,(1)}$) and "loss of effective barrier height due to blocs trapped in the nets" ($P_{4,(2)}$)

3.b) It is assumed that a correct calibration of the penalty coefficients has provided the following values:

$$P_{4,(1)} = 0.5$$

$$P_{4,(2)} = 0.85$$

Concerning the calibration of coefficient $P_{4,(2)}$, the same considerations reported previously hold, while for $P_{4,(1)}$, technical data from the real-size testing procedures and/or computer simulations of the impact of a block on the fence could provide sufficient information to assign reasonable values.

3.c) It can be considered in the first place that the penalty coefficient $P_{4,(1)}$ mostly affects the energy absorption capacity of the barrier while, in line with the idea discussed in the previous example, coefficient $P_{4,(2)}$ modifies the probability of reach.

Therefore, the reduced capacity of the barrier can be computed as:

$$E_{red} = E_{eff} \cdot P_{4,(1)} = 500 \cdot 0.5 = 250 \text{ kJ}$$

$$T_{red} = T_{eff} \cdot P_{4,(2)} = T_{eff} \cdot 0.85 = T_{opt} \cdot 0.90 \cdot 0.85$$

The energy absorption capacity of the barrier, reduced of 50%, is now lower than the reference energy value of the block at that location. If a block with an energy of 400 kJ hits the protection, it will destroy it and go through. The residual energy of the block can be estimated roughly as the difference between the energy at the impact (400 kJ) and the energy of the barrier ($E_{red}$=250 kJ), i.e. 150 kJ (approach used also in some computer simulation codes). By applying the same reduction of 50% everywhere on the slope beyond the barrier, the energy at the location of the built areas is $150 \cdot 0.5 = 75$ kJ. The new hazard level in presence of the existing barrier fence can then be estimated at this location as:

$$E(x) = 75 \text{ kJ}$$

$$T(x) = T_{red}(x) = T_{eff}(x) \cdot 0.85 = 160 \cdot 0.85 = \qquad = 136 \cong 140 \text{ years}$$

4) *Reclassification of hazard and requalification of the areas concerned in terms of land use.*

According to the results obtained, a reclassification of rock fall hazard in this case is not possible (Fig. 9). Even though the effect of the barrier fence is still to lower energy and return period at the area of interest, the point representing the new hazard still belongs to the domain of hazard marked in blue (moderate) in the Swiss matrix diagram - despite in both examples the return period decreases from moderate to low (>100 years) in presence of the barrier fence, the energy in the second example stays moderate. The protection measure in its current condition cannot therefore be taken into account for protecting the existing buildings and in view of a possible requalification in terms of land use of the area concerned; maintenance or replacement of the current barrier would be required.

**5 Discussion**

The methodology presented constitutes a tool for evaluating the role of protection measures in rock fall hazard assessment and zoning, which represents a good compromise between (i) dealing with a fairly solid amount of information to collect and use, (ii) being relatively simple in terms of application, and, at the same time, (iii) flexible.

**5.1     Rock fall protections database**

Building up a database of existing types of rock fall protections actually in use was surely a considerable amount of work to do. Even if the following estimate is rough and indicative for what concerns the effort made per protection, in terms of time and resources per protection measure and area, working just on barrier fences and dams, for instance, required in some cases time frames in the order of one-to-few days per measure, with one or two operators (but this clearly depends also on the extent

of the protection, on the number of operators available etc.; in a broader sense, estimating resources for a whole area involves defining also how large the area is, how many protections there are, which type and, again, what their extent is). In return, creating such a database helped a lot in clarifying which points require particular attention in terms of design and maintenance of each protection and, therefore, which information about a given protection should be carefully collected when they are inspected for the application of the methodology. On the other hand, despite the database constitutes a useful preliminary step

to the application of the approach proposed, it is not mandatory. Once the most important features of the protection measures to be investigated are given the appropriate consideration, in terms of quality and amount of relevant data collected, the applicability of the methodology could still be extended to territories (countries or just regions) whose surface is far greater that the surface investigated in the Canton of Vaud and for which an extensive survey aimed at building a database of the existing protections could be too demanding.

Regarding the information collected, it has to be pointed out that some protection measures are used more commonly than others, e.g. barrier fences, dams, anchors and anchored mesh wire/cable nets rather than some types of walls, re-profiling the slope, protection galleries for roads. Consequently, the amount of information available for some measures used less frequently is not as large and complete compared to the most common - this is also due to the fact that, e.g., modifying the profile of a slope or designing a gallery are complex projects which can be carried out using a number of different solutions, depending

on the conditions of the site; thus, different aspects have to be considered and, afterwards, monitored for the correct operation of the measure, depending on the specific solutions adopted. However, the database does have an evolutionary structure and can be updated as soon as new and/or more accurate information becomes available. In principle, efforts should therefore always be done to collect as relevant and detailed information as possible; should some information then be partially missing, the chief designer should use at best his engineering judgement, formulating assumptions with sufficiently good bases, and/or

try to refer to similar situations - for instance derived from similar databases (Canton of Grisons, 2017; Frei, 2013; BAFU, 2017), which might be considered as applicable also to the current study. In both cases, the results of the evaluation should be interpreted (and conclusions drawn) with particular care.

## 5.2    Considerations on the application

The procedures of data collection for examining the current conditions of existing protections and evaluation of the effectiveness of the measures are in principle quite simple.

For data collection, the forms mentioned in Section 3.2.1 already provide the basic structure of the data to be collected, and the information required can be recorded fairly easily on site during field investigations.

Regarding the evaluation, no specific detailed method has been fully developed so far for computing the reduced and effective capacities of the protections, nor to reclassify hazard based on the natural scenario and the effective capacity. On the one hand, as clarified in the introduction to this paper, this degree of detail was not (yet) sought after in this paper, which rather aims at providing a methodological framework to solve the problem of how to evaluate whether protection measures should be taken into account into hazard analyses. This feature reflects the character of the original methodology, developed for the Canton of Vaud, in which once a general framework was delineated for the approach to the problem, freedom was given to each Municipality, bureau or company to carry out the more technical parts of the evaluation with their own methods. In other words, for practical applications especially, the objective of the original methodology was not to impose a detailed methodology which would overcome the "know-how" of each professional involved in such analyses (which very often is very complex), but to provide them with a general common approach, within which each professional could work with the most appropriate methods, in the attempt of compromising at best between the consistency of the approaches used and the operators' own experience. On the other hand, consistency is crucial for the applicability of a "recommended" approach, also in view of applications for larger areas and different context than those the methodology was initially developed for. Only an appropriate degree of consistency in the operations can ensure scientifically sound and reliable results, and avoid misleading conclusions derived from the application of the methodology, should the users have too different approaches. In this sense, the procedure proposed in this paper improves the original approach proposed specifically for the Canton of Vaud. It has been mentioned that the structured method of data collection provides help in this respect already at the initial stage of the methodology (Section 3.2.1). Additionally, for what particularly regards the application, Sections 3.2.2 to 3.2.4, as well as the two examples provided, in spite of the many assumptions and simplifications made, showed some leads on how to achieve consistency. The basic idea of adopting a heuristic approach for computing the actual capacity of the measure (i.e. effective or reduced), as introduced in Sections 3.2.2 and 3.2.3, is in course of further development and is intended to be integrated in the methodology; this will provide any potential users (not only practitioners) with a slightly more pragmatic and uniform approach, compared to a general scheme of application only. In this respect, appropriate penalty coefficients will be defined for all the factors introduced, to establish which of them should rather act on the energy that the measure can retain, or the probability of occurrence, or both of them. The work in course of development aims at providing more precise elements on how these parameters should be assigned (compared for instance so some of the ideas given in Sections 4.1.1 and 4.2.1), in a way that, once validated, the resulting approach could be not only relatively fast and simple to use, in practice as well as in research, but also keep a proper scientific basis. At the same, a penalty coefficient-based approach, under the condition of

appropriate calibration of the coefficients, could still allow to integrate the experience of the engineers carrying out the analyses, for instance by optimising the choice of the penalty coefficients, so that the analysis should only benefit from relevant engineering judgement, rather than being negatively affected by it.

Concerning how to perform the reclassification of hazard, the presence of a protection measure (or, as it is more frequently done in Switzerland, a combination of them) should in principle mitigate the hazard completely, yielding a negligible hazard at the areas that the measure is meant to protect (in the Canton of Vaud in Switzerland, for instance, the tendency is to assign directly residual hazard – hatched zone in the Swiss diagram - after the protection is installed). In fact, as a function of the type of measure, e.g. a barrier fence or a dam located in the propagation zone rather than an anchored net fixed on the unstable rock wall, some events can indeed be stopped by the measure, but at the same time others, maybe even of smaller importance, can still occur and reach areas down slope. As an example, a barrier fence might be designed for mitigating an event characterised by a large volume of the blocks and therefore high energy; even when this energy is absorbed by the barrier, other events might occur soon after, and the blocks could still roll/jump over the fence filled up by the blocks of the bigger event already retained by the barrier, and propagate downslope. Although such a situation would be much less severe in terms of potential risks, it would not represent a scenario characterised by a negligible hazard, therefore it should not be neglected in any case. The same could happen, for instance, if maintenance of the barriers is not done as frequently as it should, and the barriers might be filled up and/or damaged by smaller rock fall events (some of which might even be not recorded). In the same way, if measures are taken directly on the cliff to prevent big volumes to detach, e.g. wire mesh/cable net, still smaller blocks could fall and propagate downslope, generating hazards.

Depending on the situation, the principle of reclassifying hazards should therefore take these aspects into account; in other words, from e.g. a high hazard scenario at a given location before the measure is put into place, the hazard evaluated after the installation of the measure should not necessarily be reclassified directly as residual, but it could in fact be moderate, low or residual, as a function of the degree of mitigation obtained (Fig. 10) - as shown for instance in the two examples in Sections 4.1.1 and 4.2.1.

### 5.3    Flexibility of the approach

The approach proposed presents also a certain degree of flexibility. As mentioned throughout the paper, the same procedure for evaluating the role of the protections is applicable to both existing and new protection measures, including situations in which the two could be present at the same time. An example of combination of existing and new measures could be that of a cable net already existing on the wall, which could still be effective many years after it was installed (as this kind of measure does not require a significant maintenance), and a new barrier fence, installed on the propagation zone of the rock fall, for capturing potential smaller blocks which might still fall despite the net.

Also, as it does not have any feature linked to the site or the situation for which it can be applied, the methodology can in principle be applied to any rock fall scenario and study site.

Furthermore, despite it was developed based on the Swiss guidelines for hazard zoning and the Swiss recommendations for taking into account rock fall protection measures in hazard zoning, the considerations behind the methodology are such that this approach could be applied to any other country. The methodology is indeed not dependent on the guidelines, so the evaluation of the protections' effectiveness would not be substantially influenced by the legislative constraints of other countries (besides possible norms for the correct design and maintenance of the measures). Similarly to the Swiss context, the approach could be used based on different intensity-frequency diagrams and corresponding land use planning regulations. In fact, several methodologies for rock fall hazard zoning, regional or national, are currently based on intensity-frequency diagrams (Interreg IIc, 2001; Crosta and Agliardi, 2003; Corominas et al., 2003; Jaboyedoff et al. 2005; Abbruzzese and Labiouse 2013); applying the methodology proposed internationally could be therefore quite straightforward.

## 6     Conclusions

This work presents an approach for evaluating the effectiveness of protection measures and their influence on rock fall hazard zoning. It is based on the Swiss Federal Guidelines for hazard zoning and is constituted by four steps, which include the evaluation of the conditions of existing measures and/or the choice of new ones to be installed, a two-step evaluation of their actual capacity and the possible reclassification of hazard, as a function of the mitigation role played by the measure.

The methodology features a simple structure, and can be used both for existing and new protection measures. It is based on a rock fall protection measures database which can assist the engineer in the choice of new protections to be installed and in the evaluation of the state of the current ones.

The flexibility of the methodology allows to apply it to any type of protection measure recorded in the database, for any rock fall scenario, at any site and, in principle, according to any national or regional guidelines for hazard zoning based on an intensity-frequency diagram.

*Acknowledgements.* The authors would like to thank the Authorities of the Canton of Vaud for supporting the work presented in this paper, based on the project "Falaises – FAVD", funded by the General Directorate of the Environment – Natural Hazards Unit of the Canton of Vaud.

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

| Degree of hazard | New buildings | Existing buildings |
|---|---|---|
| High | Construction of new buildings prohibited | Normal maintenance of building permitted. Modifications only when N° of persons is not increased |
| Moderate | No new residential area permitted. **In existing housings lots, construction permitted with conditions (local protection)** | **Modifications only with increased safety measures (local protection)** |
| Low | **Local protection recommended. Protection required for sensitive buildings** | No restrictions |
| Residual | **Protection required for sensitive buildings** | No restrictions |

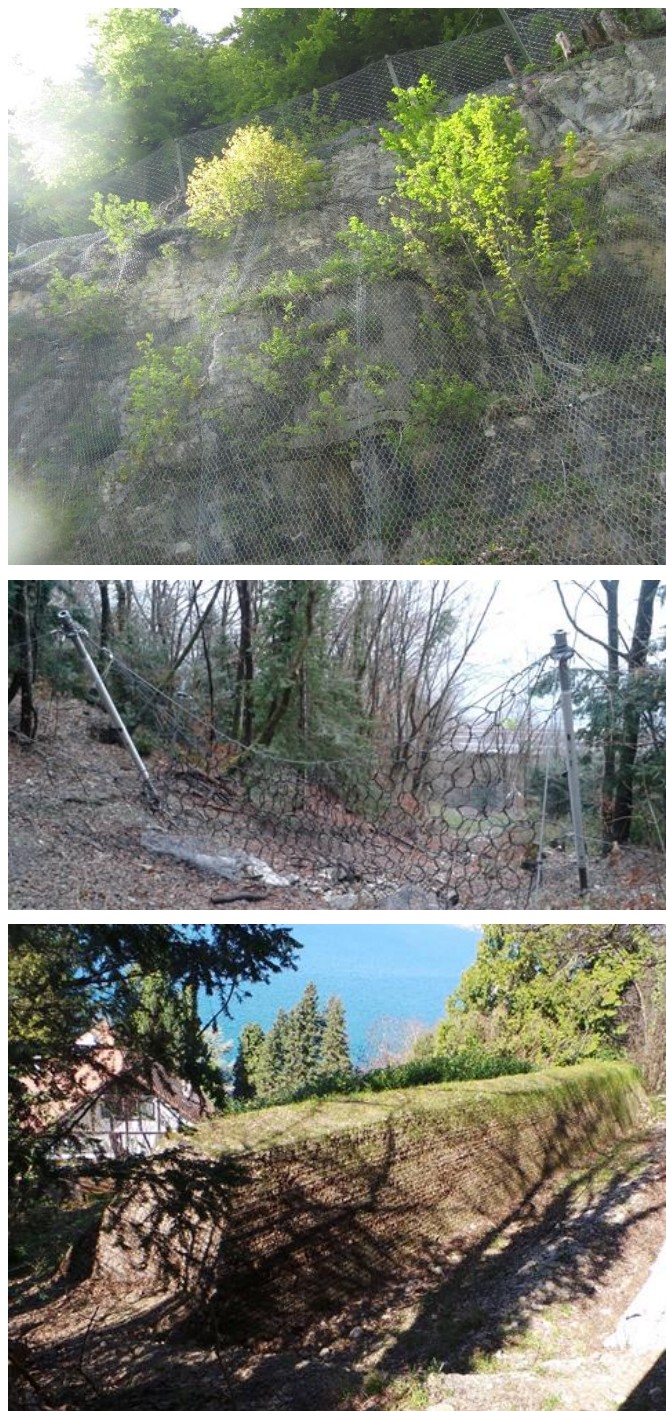

**Fig. 2. Protection measures installed on the slopes above the city of Montreux (Switzerland). Above: barrier fences and cable nets on the cliff overhanging the Cantonal Road near the town of Vallorbe. Centre: barrier fence (Source: Grisanti and Prina Howald, 2015). Below: earth dam (Source: Privat and Prina Howald, 2014).**

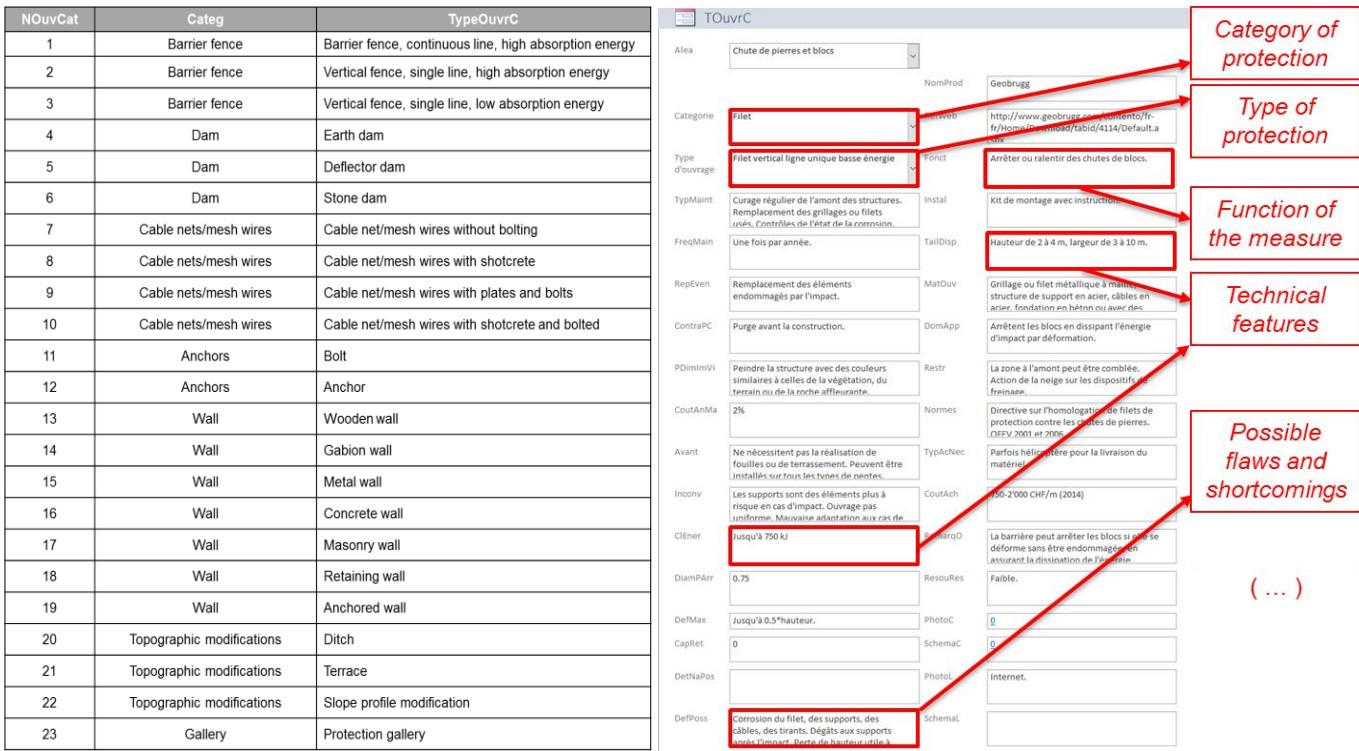

**Fig 3. Database for rock fall protection measures. Left: types and categories of rock fall protections included in the database. Right: example of form implemented in Access for each protection (here referred to a vertical barrier fence installed in single line, with low energy retention capacity).**

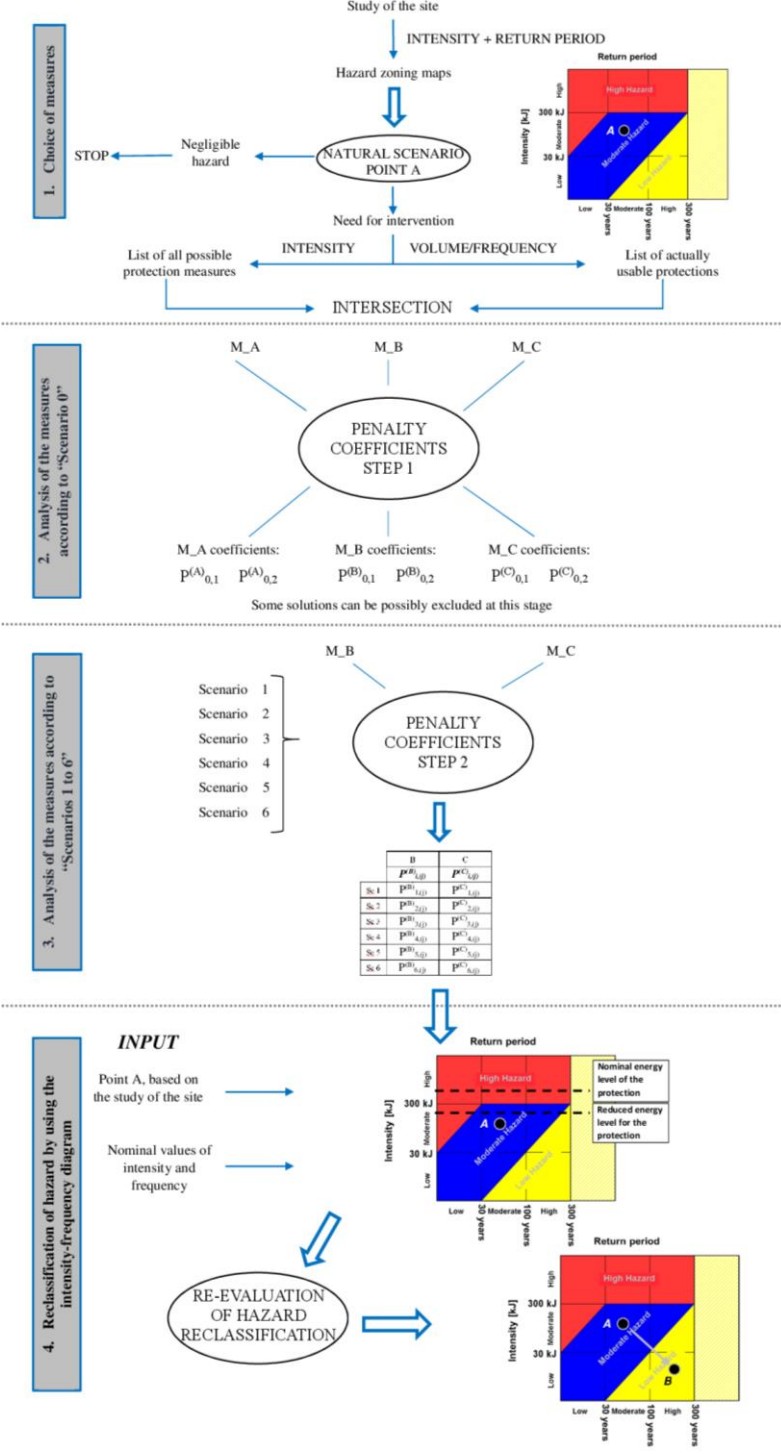

**Fig. 4. Scheme of the methodology for evaluating the effectiveness of rock fall protections and reclassifying hazards (modified from Grisanti and Prina Howald, 2014, 2015a and 2015b).**

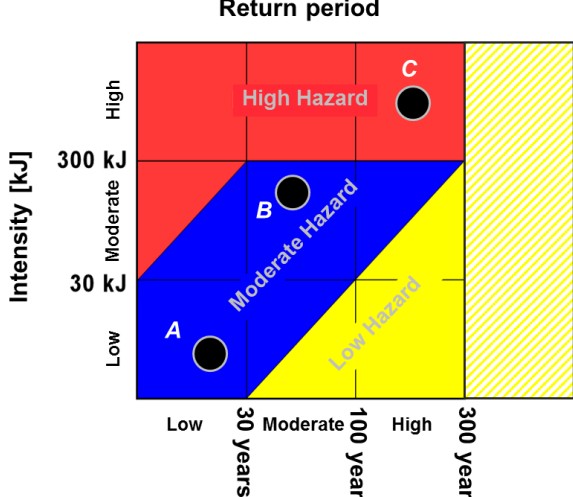

**Fig. 5. Characterisation of the initial hazard at a given site. (A), (B), (C): examples of possible scenarios for low, moderate and high hazard, respectively, at a given location on the slope.**

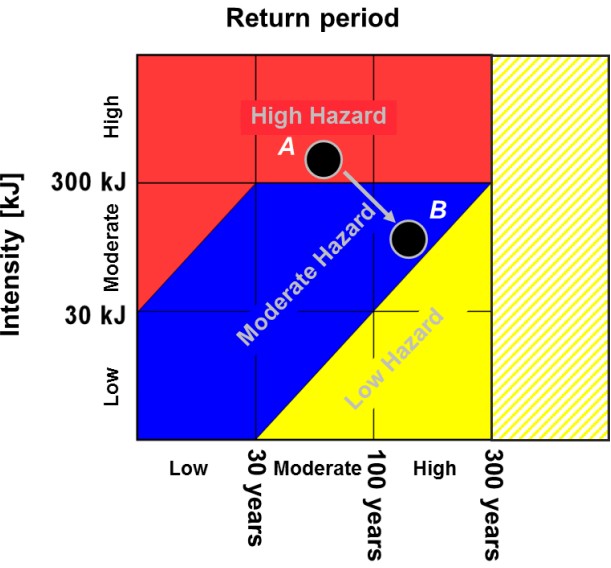

**Fig. 6. Reclassification of hazard at a given location of the slope, using the Swiss intensity-frequency diagram. Point A: natural rock fall hazard scenario; Point B: new hazard level assessed once the effect of a given protection measure is taken into account.**

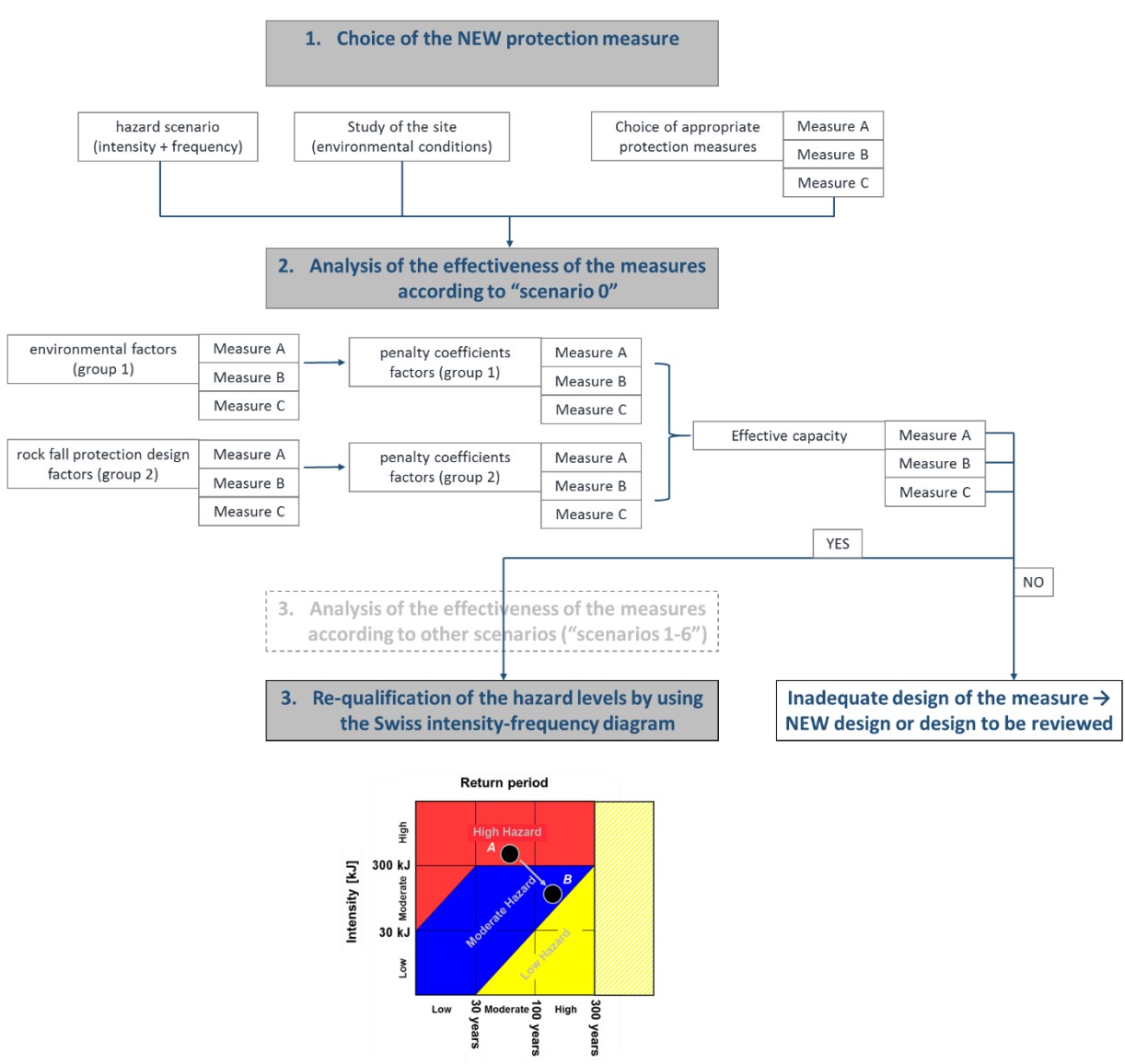

**Fig. 7. Flow-chart for the application of the methodology for new protection measures.**

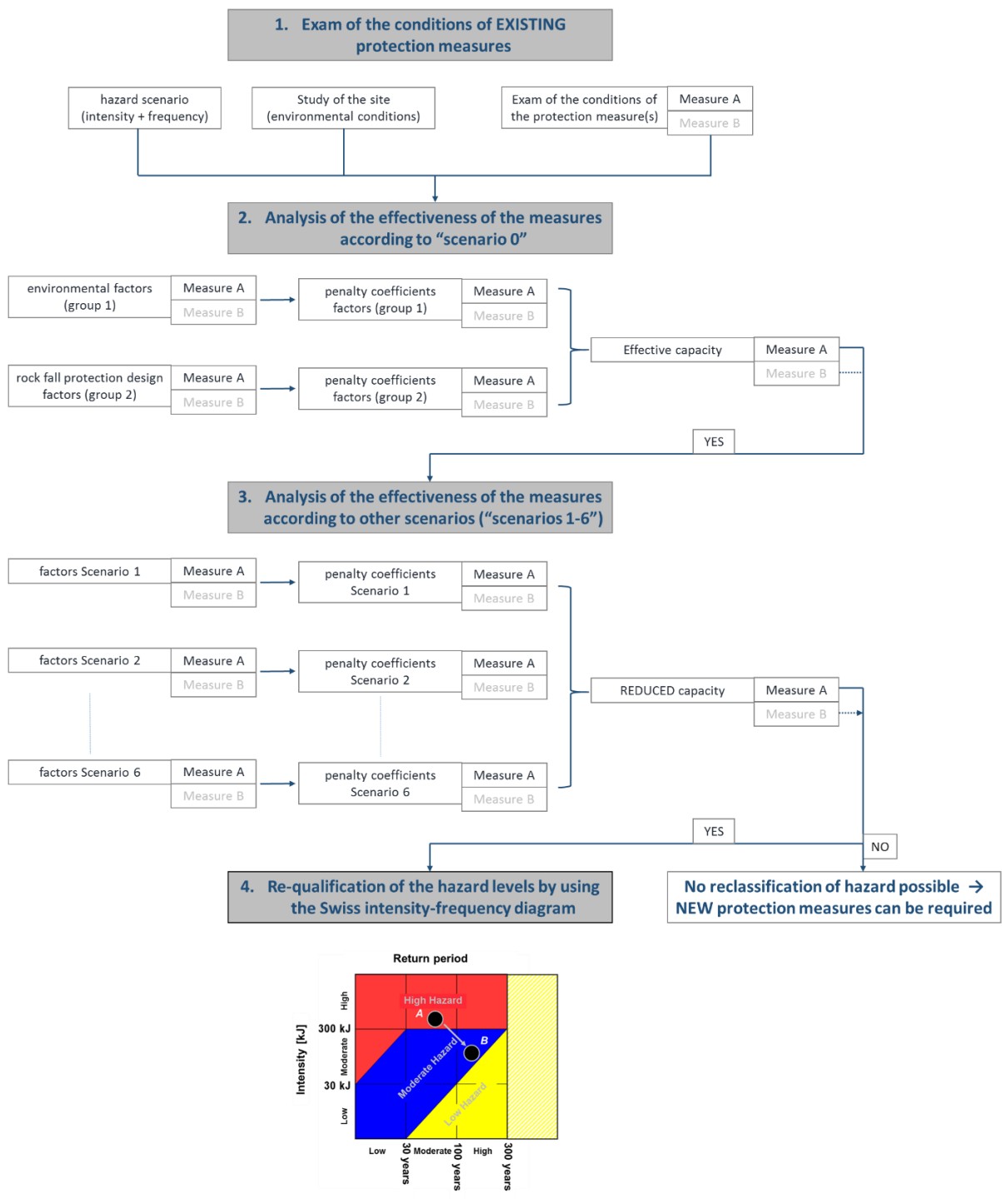

**Fig. 8. Flow-chart for the application of the methodology for existing protection measures (options written in light grey and dotted arrows are referred to the possibility of taking into account more than one protection already existing at a study site).**

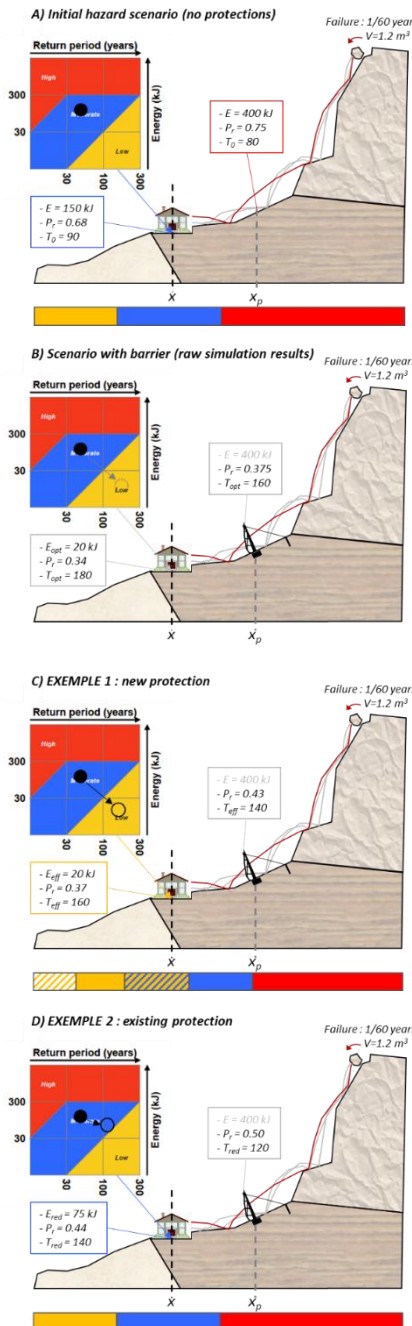

**Fig. 9. Example of application of the methodology. A) Hazard scenario before protections; B) theoretical hazard scenario obtained in presence of the protection based on raw rock fall simulation results; C) Methodology applied for new protections: reclassification of hazard possible; D) Methodology applied for existing protections: reclassification of hazard not possible. Below each scheme the corresponding hazard zoning is reported: for cases C) and D) this is obtained by applying the methodology to every abscissa beyond**
10  **the rock fall barrier.**

**Fig. 10.** Example of possible hazard reclassification at a given location of the slope, starting from a natural scenario characterised by high hazard.