# Peer review of "An approach for evaluating the role of protection measures in rock fall hazard zoning based on the Swiss experience"

_Natural Hazards and Earth System Sciences, 2016_

## Referee Comment (RC1) · Anonymous Referee #1 · 29 Sep 2016

The paper presents a procedure for evaluating the actual performance of the risk mitigation measures and how it could affect the assigned hazard level of the protected areas. The methodology considers two steps that take into account various factors that may interfere on the performance of the measures. Penalty coefficients are applied in case the efficacy of the measures is adversely affected by the considered factors. The paper is well written and the topic certainly fits within the scope of NHESS. The approach is novel and addresses a gap in the rockfall hazard analysis which is the consideration of the protected areas. The authors must be acknowledged for this effort. In my opinion, the rationale of the methodology must be better supported and the paper needs further developments before it could be accepted.

[Figure]

I found three main issues should be specifically addressed and require in depth consideration:

1) The authors state that the procedure is straightforward. However, it is unclear how the methodology is applied in real cases. In particular, how the resultant effective capacity is integrated in the calculation of the intensity-frequency value. A worked example should be presented to illustrate it. 2) I cannot see the reasons that justify the use of this procedure for Scenario 0 or the design of new mitigation measures. Engineering designs such as the rockfall stabilization and protection works usually follow to the specifications of the design codes prepared by professional societies and/or administrations. Generally, the chief designer must prove that the structure satisfies the specified performance requirements, such as the safety and serviceability with the appropriate levels of reliability and that he/she has followed the procedures defined in the codes for the design calculation. If environmental factors (p.6 lines 14 to 17) affecting the efficacy of the structure are identified, this must be considered in the engineering design as well. Even though some remedial works might not be safe enough in practice, the underlying concept in the methodology that their design is independent of the context of the site is not appropriate for a recommended procedure. 3) The last point is the concept that hazard can be reevaluated. The remedial measure is an acceptable option in case of an existing risk (presence of exposed elements in a threatened area). It is clear that the land beyond the protection structure is safer as the probability of occurrence of the damaging event has been significantly reduced. However, the decision of changing the hazard level is not evident. A low hazard and risk level due to the absence of the threat has a different meaning than the risk being reduced by the existence of a protection work. In the latter, the threat is still there and the hazard zonation proposed by the authors is closely linked to the expected performance of the protection work. If as mentioned in the abstract, the procedure aims to be extended elsewhere, the criteria for hazard reclassification must be well defined and in the paper it is not.

Minor comments:

Page 2. Lines 4-6. Land use regulations aim at avoiding exposure to the hazard rather than reducing the consequences. The latter are usually the goal of the protection measures.

Page 3. Lines 10-14. Residual risk is usually interpreted as the one remaining after all efforts to mitigate risk have been made. You should add a sentence highlighting the difference with the concept of "residual hazard" used in Switzerland and in the paper.

---

## Referee Comment (RC2) · Anonymous Referee #2 · 29 Sep 2016

The presented article provides a framework on how to standardize the evaluation of protection measures against rockfall. This evaluation then further can be used to evaluate its hazard/risk mitigation.

1) For the application of the procedure I see one major issue: The article clearly states that the framework presented does not contain specific details on how to for example choose the values for the penalty factor. I.e., such decisions are to be made by the operator. The problem now is on how to guarantee a consistency of the database's contents and how to avoid individual influences originating from different operators. Only, if this consistency is guaranteed the database will be usable also regarding a larger area.

[Figure]

2) The article could also setup some ideas on how to achieve above demanded consistency. Further, some rough estimations on the necessary effort per protection measure and per area could be included. The experience resp. the data due to the existing data retrieval in Vaud probably exist.

3) The article could also rely on existing similar databases regarding an inclusion into the presented method. E.g. the canton of Graubünden has an inventory on their protection measures: http://map.geo.gr.ch/verbauungen and http://www.mfrei-infra.ch/cms/fileadmin/pdf/SchutzbauManagement2013.pdf The Swiss BAFU e.g. recommends the so-called KUFI-Handbook http://www.bafu.admin.ch/wirtschaft/15300/15310/15316/index.html

4) What happens if the operator cannot answer detailed/important information on certain protection measures?

Short correction page 4 line 20: "was" –> "is".

---

## Author Comment (AC1) · 13 Dec 2016

Thank you for your valuable comments to our paper. Please find here below our answers to your comments and suggestions.

1) The idea of an example illustrating the application of the methodology is surely valuable and relevant; we definitely agree it would in principle provide a better insight on the application, step by step. However, an example could not be included in this paper for the following reasons. As specified in the paper (Page 2, lines 25-27), the methodology derived from a rather "specific" study we carried out for one region (Canton of Vaud) in one country (Switzerland). Its main goal was primarily to present a methodological framework, within which many aspects and issues characterising the evaluation of haz-

[Figure]

ard in presence of protection measures could be structured into steps to be followed by professionals and firms (Page 13, lines 6-16), in order to have a clearer view of the problem and on how to possibly solve it. The approach proved itself interesting to be potentially extended in a broader sense, i.e. at the international level, rather than regional or national, and involving both the practitioners and the research community. Therefore, efforts were focused primarily on extending the methodological framework, at first. At the same time, we are indeed very much interested in going deeper into this work and give more operational and quantitative solutions to solve the most technical aspects of the methodology within this framework we proposed. The application/testing phase of the methodology has actually been started, but its developments are still in the very beginning, including an appropriate calibration of the penalty factors (which has still to be done, see Page 13, lines 19-26), a sufficient number of test sites to be defined for validating the methodology, as well as a suitable method for using the reduced capacity of the protection to estimate a residual hazard (energy-return period couple on the intensity-frequency diagram). This is why it was chosen to focus mainly on the general framework of the methodology at this stage, rather than showing an example which would not have not been fully relevant and meaningful, due to too uncertain values assigned to the penalty coefficients, as well as too rough estimates of the residual hazard as a function of the reduced capacity. The straightforward applicability of the methodology mentioned in the paper referred for the moment to the application of the methodology based on an intensity-frequency matrix different than the Swiss, which is indeed possible without any modification of the approach. In other words, the computations to be done to estimate residual hazard are not based on the intensity-frequency diagram used (the diagram is used only to check which new hazard level is obtained after these computations).

2) For already existing protections: the use of Scenario 0 is justified by the fact that environmental conditions and/or even norms and directives concerning the design of protection measures can change in time, even within the life span of a protection. If this is the case, when the state of the measure is evaluated after a certain time it was installed, the potential influence of the factors considered in Scenario 0 should be taken into account (e.g.: some recommendations/norms might have partially changed after, for instance, a specific event, and because of this some technical specifications of the measure might require to be adjusted; damages due to animals might have occurred even when not expected; after a few minor events occurred since the measure was installed, its possibility of plastic deformations has been reduced, etc.) For the design of new measures: the observation that all factors related to Scenario 1) should already be accounted for in the designing codes and respected by the chief designer is absolutely correct, we do agree (and we do agree that this is valid for existing measures as well). Scenario 0, basically, simply allows to evaluate the effects of those factors which are/should be existing in the designing codes (and does not aim at "substituting" designing codes), for possibly mitigating them and ensuring that the protection can work according to its nominal capacity (as it should, since it is newly designed). Capacity and position of a protection are often determined based on rock fall simulations, and most of the factors in Scenario 0 cannot be simulated; thus, the methodology aims at recollecting these factors and establishing how they affect the nominal capacity determined from simulations results. This evaluation is relevant, as engineering solutions to mitigate the potential influence of the factors belonging to Scenario 0 should be based on the specific protection chosen; Scenario 0 helps in detecting whether the reduction of capacity for the chosen protection is significant or, despite some environmental factors might play a role, negligible (i.e. the protection chosen can still work perfectly and no adjustment/new design is needed). Based on these considerations, therefore, the methodology does not aim at designing the measures independently from the site conditions, but rather the contrary, and this is the reason why environmental factors and factors somehow linked to the interaction between the site and the protection are taken into account.

3) Thank you very much for another very constructive comment: again, we fully agree with these considerations. Indeed, the situation in which a given hazard level is solely due to a "natural" scenario is different from the situation in which the same level of

hazard is the result of a mitigation obtained relying on the performance of a protection. In fact, in the methodology proposed, the reclassification is planned to be done by representing the new hazard level with a hatched zone in all the area(s) concerned of the hazard zoning map. This hatched zone will feature both the colour corresponding to the hazard level associated to the event ("natural" scenario), and the colour corresponding to the new hazard level obtained in presence of protections ("final" scenario). The aim of this representation is exactly to inform users that the new (lower) hazard level at the area(s) concerned is the one resulting from the final scenario, because a protection actually is working to reduce the hazard (which would otherwise be higher, i.e. other colour in the hatched zone). We reckon this should have been included in the paper. We propose to include these details in Section 3.2.4.

Minor comments:

Page 2. Lines 4-6. Thank you very much for the comment, we will make corrections accordingly.

Page 3. Lines 10-14. We propose to modify the following definitions, according to your suggestion: Page 3. Lines 10-14 : "residual hazard" (Switzerland): "An additional hazard level qualified as "residual" is defined for return periods higher than 300 years; this definition is based only on the probability of occurrence and identifies the possible occurrence of extreme rare events (but no further specification or well defined classes are given in terms of return period and energy to classify this hazard). In the paper, the concept of residual hazard is used with the same meaning, as it always used in relation to the corresponding hazard zone defined in the Swiss Codes. We could precise though that, even according to the Swiss Codes, if residual hazard is the result of the performance of a protection measures, the event should not only be considered as "rare" in the area concerned (very low probability of occurrence), but the definition somehow implies also a very low intensity (as most measures work on the energy of the event too) – on the contrary, if no measure is in place, residual hazard is associated to events which have a very low probability of occurrence also because of their

extremely high and rare magnitude.

---

## Author Comment (AC2) · 13 Dec 2016

Thank you for your valuable comments to our paper. Please find here below our answers to your comments and suggestions.

1) It is true that, at the current stage, the methodology has been designed to leave quite some freedom to the practitioners to implement their know-how into the evaluation of the reduced capacity of the protections and the residual hazard. However, this freedom should not affect significantly the values to be given to the same penalty coefficients by two different users (it could on the other hand concern some other aspects, e.g. the structure of the equations used in the heuristic approach - i.e. some coefficients might be weighted - or similar ideas. . .). We are fully aware that consistency is very important

for ensuring scientifically sound and reliable results, and avoid misleading conclusions derived from the application of the methodology, should the users have very different approaches. It is for this reason that we proposed in the paper (Page 13, lines 16-26) the use of one given approach (e.g. heuristic) for the application of the penalty factors and, in particular, specified that "appropriate penalty coefficients should be defined for all the factors introduced" (Page 13, line 19) – so that if, for instance, a range of values suitable for most applications is defined for each penalty coefficient, all users can use values of the coefficients taken from that range (and only in special and well justified cases opt for more extreme values). We want to point out, at the same time, that also the experience and knowledge of the engineer/chief designer who is leading the study is important, however, and should play a positive role in the evaluation. Once the operating framework is indeed as uniform and consistent as possible, the results of the analysis should only benefit from his appropriate engineering judgement, rather than being negatively affected by it.

2) Regarding the consistency of the approach, some leads were given in the previous answer (heuristic approach, range of suitable values for most applications for each penalty coefficient) and in the paper (Page 13, lines 16-26). For what concerns the effort per protection, it is indeed very complicated to give even a rough estimation of the investment in terms of time and resources per protection measure and area. Working on barrier fences and dams required in some cases time frames in the order of one-to-few days per measure, with one or two operators - but this depends also on the extent of the protection, on the number of operators available etc. Estimating resources for an area is even more complicated, as it involves defining how large the area is, how many protections there are, which type, what their extent is (for instance for barrier fences and dams), etc. This information might be hard to retrieve in some cases for existing sites already studied, as well as hard to project to areas which have to be studied next.

3) Thank you very much, we will surely consider the inclusion of similar databases and

we will mention in the paper the references you suggested. On the one hand, this proposition is very useful to expand and complete the database itself and, ultimately, improving its quality with new and/or more detailed data about the type of measures and their specifications. On the other hand, such an inclusion would reflect a different experience to be merged in the methodology, in comparison to that existing in the Canton of Vaud, thus broadening the character of the methodology. This could consequently represent a first step in making it more applicable at the national and, with possible further data deriving from other databases built up abroad, internationally.

4) This issue is surely relevant, as basically all the steps of the methodology are clearly linked to the amount and quality of the information which is possible to collect for the protection measures at the beginning of the procedure. As for every aspect of any methodology, efforts should be done to collect as relevant and detailed information as possible. However, should required information still be partially (or, even worse, totally) missing, the chief designer should use at best his engineering judgement, formulating assumptions with sufficiently good bases, and/or try to refer to similar situations, for instance derived from similar databases, which might be considered as applicable also to the current study. In both cases, the results of the evaluation should be interpreted (and conclusions drawn) with particular care.

Minor comments:

Short correction page 4 line 20: "was" –> "is". We will correct as suggested, thank you.
* * *

---

## Author Response (AR1)

**Answers to interactive comments**

Thank you for your valuable comments to our paper. Please find here below our answers to your comments and suggestions.

*Referee 1:*

**1)** The authors state that the procedure is straightforward. However, it is unclear how the methodology is applied in real cases. In particular, how the resultant effective capacity is integrated in the calculation of the intensity-frequency value. A worked example should be presented to illustrate it.

*Authors' reply:*

*Thank you for your suggestion: we included two examples in Sections 4.1.1 and 4.2.1 for the application of the methodology for new measures and existing measures, respectively*

*Referee 1:*

**2)** I cannot see the reasons that justify the use of this procedure for Scenario 0 or the design of new mitigation measures. Engineering designs such as the rockfall stabilization and protection works usually follow to the specifications of the design codes prepared by professional societies and/or administrations. Generally, the chief designer must prove that the structure satisfies the specified performance requirements, such as the safety and serviceability with the appropriate levels of reliability and that he/she has followed the procedures defined in the codes for the design calculation. If environmental factors (p.6 lines 14 to 17) affecting the efficacy of the structure are identified, this must be considered in the engineering design as well. Even though some remedial works might not be safe enough in practice, the underlying concept in the methodology that their design is independent of the context of the site is not appropriate for a recommended procedure.

*Authors' reply:*

*Thank you for your remarks: example 1 in Section 4.1.1 gives some details about the application of the methodology for Scenario 0. We do agree that an appropriate design should guarantee safety and serviceability and reliability of the measure, and this is also what we decided to include explicitly in Section 4.1.1 (p.12 lines 9-11), where we present the application of Scenario 0 . In addition, we could briefly give some more details about this phase of the methodology, as follows:*

*For already existing protections: the use of Scenario 0 is justified by the fact that environmental conditions and/or even norms and directives concerning the design of protection measures can change in time, even within the life span of a protection. If this is the case, when the state of the measure is*

*evaluated after a certain time it was installed, the potential influence of the factors considered in Scenario 0 should be taken into account (e.g.: some recommendations/norms might have partially changed after, for instance, a specific event, and because of this some technical specifications of the measure might require to be adjusted; damages due to animals might have occurred even when not expected; after a few minor events occurred since the measure was installed, its possibility of plastic deformations has been reduced, etc.)*

*For the design of new measures: the observation that all factors related to Scenario 1) should already be accounted for in the designing codes and respected by the chief designer is absolutely correct, we do agree (and we do agree that this is valid for existing measures as well). Scenario 0, basically, simply allows to evaluate the effects of those factors which are/should be existing in the designing codes (and does not aim at "substituting" designing codes), for possibly mitigating them and ensuring that the protection can work according to its nominal capacity (as it should, since it is newly designed). Capacity and position of a protection are often determined based on rock fall simulations, and most of the factors in Scenario 0 cannot be simulated; thus, the methodology aims at recollecting these factors and establishing how they affect the nominal capacity determined from simulations results. This evaluation is relevant, as engineering solutions to mitigate the potential influence of the factors belonging to Scenario 0 should be based on the specific protection chosen; Scenario 0 helps in detecting whether the reduction of capacity for the chosen protection is significant or, despite some environmental factors might play a role, negligible (i.e. the protection chosen can still work perfectly and no adjustment/new design is needed).*

*Based on these considerations, therefore, the methodology does not aim at designing the measures independently from the site conditions, but rather the contrary, and this is the reason why environmental factors and factors somehow linked to the interaction between the site and the protection are taken into account.*

*Referee 1:*

**3)** The last point is the concept that hazard can be reevaluated. The remedial measure is an acceptable option in case of an existing risk (presence of exposed elements in a threatened area). It is clear that the land beyond the protection structure is safer as the probability of occurrence of the damaging event has been significantly reduced. However, the decision of changing the hazard level is not evident. A low hazard and risk level due to the absence of the threat has a different meaning than the risk being reduced by the existence of a protection work. In the latter, the threat is still there and the hazard zonation proposed by the authors is closely linked to the expected performance of the protection work. If as mentioned in the abstract, the procedure aims to be extended elsewhere, the criteria for hazard reclassification must be well defined and in the paper it is not.

*Authors' reply:*

*Thank you very much also for this comment: we fully agree with this too and, following your suggestion, we explained how the reclassification is done in Section 3.2.4 (p. 9, lines 30-33, p.10, lines 1-5).*

**Minor comments:**

Page 2. Lines 4-6. Land use regulations aim at avoiding exposure to the hazard rather than reducing the consequences. The latter are usually the goal of the protection measures.

*Authors' reply:*

*Thank you very much for the comment, we made corrections accordingly.*

Page 3. Lines 10-14. Residual risk is usually interpreted as the one remaining after all efforts to mitigate risk have been made. You should add a sentence highlighting the difference with the concept of "residual hazard" used in Switzerland and in the paper.

*Authors' reply:*

*Thank you for the suggestion, we modified the text accordingly (Section 2, p. 3. Lines 10-15)*

Nat. Hazards Earth Syst. Sci. Discuss.,
doi:10.5194/nhess-2016-265-RC1, 2016
**Anonymous Referee #2**

**Answers to interactive comments**

Thank you for your valuable comments to our paper. Please find here below our answers to your comments and suggestions.

*Referee 2:*

**1)** For the application of the procedure I see one major issue: The article clearly states that the framework presented does not contain specific details on how to for example choose the values for the penalty factor. I.e., such decisions are to be made by the operator. The problem now is on how to guarantee a consistency of the database's contents and how to avoid individual influences originating from different operators. Only, if this consistency is guaranteed the database will be usable also regarding a larger area.

*Authors' reply:*

Thank you very much for this comment, we completely agree with it and we reformulated the text accordingly (*Section 5.2, p. 18, lines 22-31*) to better clarify this same view we have on this topic.

*Referee 2:*

**2)** The article could also setup some ideas on how to achieve above demanded consistency. Further, some rough estimations on the necessary effort per protection measure and per area could be included. The experience resp. the data due to the existing data retrieval in Vaud probably exist.

*Authors' reply:*

*Thank you for this suggestion: we modified/reformulated the text for addressing both requests in Section 5.2 (p. 18, lines 27-33 + p. 19, lines 1-11, and p.17 lines 11-15, respectively).*

*Referee 2:*

**3)** The article could also rely on existing similar databases regarding an inclusion into the presented method. E.g. the canton of Graubünden has an inventory on their protection measures: http://map.geo.gr.ch/verbauungen and

http://www.mfrei-infra.ch/cms/fileadmin/pdf/SchutzbauManagement2013.pdf

The Swiss BAFU e.g. recommends the so-called KUFI-Handbook:

http://www.bafu.admin.ch/wirtschaft/15300/15310/15316/index.html

*Authors' reply:*

*Thank you very much, we will surely consider the inclusion of similar databases and we referred to them in the paper (p.5, line 4).*

*On the one hand, this proposition is very useful to expand and complete the database itself and, ultimately, improving its quality with new and/or more detailed data about the type of measures and their specifications. On the other hand, such an inclusion would reflect a different experience to be merged in the methodology, in comparison to that existing in the Canton of Vaud, thus broadening the character of the methodology. This could consequently represent a first step in making it more applicable at the national and, with possible further data deriving from other databases built up abroad, internationally.*

*Referee 2:*

**4)** What happens if the operator cannot answer detailed/important information on certain protection measures?

*Authors' reply:*

*Thanks a lot for the remark: the issue has now been addressed (p. 18, lines 1-6).*

**Minor comments:**

Short correction page 4 line 20: "was" –> "is".

*Authors' reply:*

*We corrected as suggested, thank you.*

[revised manuscript text omitted]

[Figure]

| NOuvCat | Categ | TypeOuvrC |
|---------|-------|-----------|
| 1 | Barrier fence | Barrier fence, continuous line, high absorption energy |
| 2 | Barrier fence | Vertical fence, single line, high absorption energy |
| 3 | Barrier fence | Vertical fence, single line, low absorption energy |
| 4 | Dam | Earth dam |
| 5 | Dam | Deflector dam |
| 6 | Dam | Stone dam |
| 7 | Cable nets/mesh wires | Cable net/mesh wires without bolting |
| 8 | Cable nets/mesh wires | Cable net/mesh wires with shotcrete |
| 9 | Cable nets/mesh wires | Cable net/mesh wires with plates and bolts |
| 10 | Cable nets/mesh wires | Cable net/mesh wires with shotcrete and bolted |
| 11 | Anchors | Bolt |
| 12 | Anchors | Anchor |
| 13 | Wall | Wooden wall |
| 14 | Wall | Gabion wall |
| 15 | Wall | Metal wall |
| 16 | Wall | Concrete wall |
| 17 | Wall | Masonry wall |
| 18 | Wall | Retaining wall |
| 19 | Wall | Anchored wall |
| 20 | Topographic modifications | Ditch |
| 21 | Topographic modifications | Terrace |
| 22 | Topographic modifications | Slope profile modification |
| 23 | Gallery | Protection gallery |

Category of protection

Type of protection

Function of the measure

Technical features

Possible flaws and shortcomings

( … )

**Fig 3. Database for rock fall protection measures. Left: types and categories of rock fall protections included in the database. Right: example of form implemented in Access for each protection (here referred to a vertical barrier fence installed in single line, with low energy retention capacity).**

[Figure]

**Fig. 4. Scheme of the methodology for evaluating the effectiveness of rock fall protections and reclassifying hazards (modified from Grisanti and Prina Howald, 2014, 2015a and 2015b).**

[Figure]

**Fig. 5.** Characterisation of the initial hazard at a given site. (A), (B), (C): examples of possible scenarios for low, moderate and high hazard, respectively, at a given location on the slope.

[Figure]

**Fig. 6.** Reclassification of hazard at a given location of the slope, using the Swiss intensity-frequency diagram. Point A: natural rock fall hazard scenario; Point B: new hazard level assessed once the effect of a given protection measure is taken into account.

[Figure]

**Fig. 7. Flow-chart for the application of the methodology for new protection measures.**

[Figure]

**Fig. 8. Flow-chart for the application of the methodology for existing protection measures (options written in light grey and dotted arrows are referred to the possibility of taking into account more than one protection already existing at a study site).**

[Figure]

5   **Fig. 9. Example of application of the methodology. A) Hazard scenario before protections; B) theoretical hazard scenario obtained in presence of the protection based on raw rock fall simulation results; C) Methodology applied for new protections: reclassification of hazard possible; D) Methodology applied for existing protections: reclassification of hazard not possible. Below each scheme the corresponding hazard zoning is reported: for cases C) and D) this is obtained by applying the methodology to every abscissa beyond the rock fall barrier.**

[Figure]

Fig. 109. Example of possible hazard reclassification at a given location of the slope, starting from a natural scenario characterised by high hazard.

